# Comparison and Optimization of Continuous Flow Reactors for Aerobic Granule Sludge Cultivation from the Perspective of Hydrodynamic Behavior

**DOI:** 10.3390/ijerph19148306

**Published:** 2022-07-07

**Authors:** Xinye Jiang, Hongli Li, Qingyu Zhao, Peng Yang, Ming Zeng, Du Guo, Zhiqiang Fu, Linlin Hao, Nan Wu

**Affiliations:** 1College of Marine and Environmental Sciences, Tianjin University of Science &Technology, Tianjin 300457, China; jiangxinye@mail.tust.edu.cn (X.J.); lhl@mail.tust.edu.cn (H.L.); zhaoqy@mail.tust.edu.cn (Q.Z.); 19827918@mail.tust.edu.cn (D.G.); xilan.2009@tust.edu.cn (L.H.); 2Agro-Environmental Protection Institute, Ministry of Agriculture, Tianjin 300191, China; 3School of Light Industry Science and Engineering, Tianjin University of Science &Technology, Tianjin 300457, China; zqfu@tust.edu.cn; 4College of Engineering and Technology, Tianjin Agricultural University, Tianjin 300384, China; nwu@tjau.edu.cn

**Keywords:** aquaculture wastewater, aerobic granule sludge, continuous flow reactors, CFD, hydrodynamic behavior

## Abstract

Improving treatment efficiency and reducing investment and operating costs make aerobic granular sludge technology (AGS) a promising technology for treating aquaculture wastewater. The development of continuous flow reactors (CFRs) has become a new direction in the research of AGS. This study clarifies the granulation effect, hydrodynamic behavior and particle separation of three different CFRs (R1 to R3). The established CFD model was able to explain the hydrodynamic behavior in all three CFRs; in particular, R3 performed the best from the perspective of hydrodynamic behavior due to its abundant turbulence. In addition, the optimal baffle distance and baffle angle of R3 were simulated to be 40 mm and 60°, respectively, due to them providing the best turbulent flow and particle separation effect. However, an overlarge baffle angle could weaken the turbulent pattern in the reactor. The retention time distribution further confirmed the reasonability of these optimal parameters with the highest effective volume ratio of 0.82. In short, this study gives an instruction for exploring the rapid formation mechanism of AGS in a CFR to promote its engineering application.

## 1. Introduction

The treatment of aquaculture wastewater faces a great challenge due to its high concentration of organic matter, ammonia nitrogen, suspended solids and phosphorus. The activated sludge method is an effective method for treating aquaculture wastewater by using aerobic microorganisms. The recent research has set aerobic granular sludge (AGS) as a promising technique to improve treatment efficiency and reduce investment and operating costs [1,2,3]. Its advantages over conventional activated sludge systems include: its good performance and low cost (investment and operations) by removing organic loads and nutrients, as well as the ability to treat wastewater with high organic loads or toxic substrates [4,5,6]. Othman et al. [7] established a sequencing batch reactor to treat livestock wastewater with maximum COD, TN and TP removal efficiencies of 74%, 73% and 70%, respectively.

A short granulation time is preferred because accelerating the granulation of aerobic granular sludge allows it to react with pollutants in a short period of time. Xin et al. [8] accelerated the granulation of AGS by adding calcium ions to the AGS to achieve good treatment effects in 40 days; however, it needs 72 days without calcium ions.

The development of continuous flow reactors has become a new direction in the research of AGS [9]. Continuous flow reactors (CFRs) have become a research hotspot in recent years because of their advantages such as their easy operation and control, large amount of water treatment, low operating cost and small occupation zone. This technology requires the aerobic granules to be kept stably in CFRs [10]. However, it is difficult to keep aerobic granules held in CFRs systems [11]. Therefore, exploring the rapid formation mechanism of AGS in a CFR is of great significance to promote its engineering application.

At present, there are only field-scale studies of AGS in CFRs, with no field-scale studies of CFD on CFRs. Computational fluid dynamics (CFD) is a numerical method to address fluid flow and heat transfer and other related physical and biochemical processes, which has been widely and successfully used in the study of wastewater treatment systems [12,13,14,15,16]. In recent years, many studies have shown that CFD can be applied to the design of bioreactors. Mutharasu et al. [17] used a CFD model to evaluate the hydrodynamics of the novel down-flow bubble column. Wang et al. [18] studied the influence of a water distribution system on the flow field inside the reactor. The shortcomings of the reactor can be revealed and optimized, such as velocity inhomogeneity and the stagnant zone [19].

This study aimed to investigate the use of computational fluid dynamics (CFD) to optimize the generation and stable operation of aerobic granular sludge (AGS) in a continuous flow reactor (CFR). The study utilizes the data presented in three previously published papers [6,8,20] to generate the CFD models. The objectives include: (1) comparing the granulation efficiency of three CFRs; (2) simulating the hydrodynamic behavior of granules to fix an optimal CFR; (3) optimizing the structure of the selected CFRs. This study helps to understand the feasibility of the use of CFRs for the formation and separation of AGS, and provides a new technical solution for aquaculture wastewater treatment.

## 2. Materials and Methods

### 2.1. Bioreactors of CFR

Different operational conditions might determine the efficiency of granulating process. The operational parameters of three CFRs are summarized in Table 1, which can be divided into three types: reactor structure, water and gas conditions. All three reactors belonged to the lab-scale ones and R3 owned the largest volume of 10 L.

According to the references [6,8,20], the hydraulic residence times of the reactors were 0.55, 0.90 and 10.00 h for R1, R2 and R3, respectively. R3 had the lowest water flow rate and highest air flow rate, and, conversely, R1 owned the highest water flow rate and lowest air flow rate. The granular diameters of the reactors were 0.5–2.0, 1.2 and 2 mm for R1, R2 and R3, respectively.

Regarding the granulating efficiency, there is a difference in the granulation time of the AGS in the three CFRs (Table 1). The round or oval brown granular sludge with a size of 0.5–2.0 mm was observed in R1 after 40 days. The SVI_5_ decreased from the initial 122.62 to 46.61 mL/g, and then remained at 44.28–60.51 mL/g. The biomass concentration (in terms of Mix Liquor Suspended Solids) was 3.67 g/L. The nitridation granular sludge in R2 was brown-red, compact and circular, with an average particle size of 0.9 mm. Seed granular sludge was cultivated in a lab-scale SBR with a 4.0 L working volume, which had been operated at a nitrogen loading rate (NLR) of 1.0–1.5 kg N/m^3^/d for more than 160 days. The biomass concentration (in terms of Mix Liquor Suspended Solids) was 0.84 g/L with an SVI_5_ of 20 mL/g after 33 days’ cultivation. Finally, the shortest granulation time belonged to R3, where a group of particles with a particle size between 0.1–0.5 mm began to appear after only 2 weeks of cultivation. Two liters of sedimentation sludge with a sludge volume index of 40 mL/g was added in the continuous flow reactor as the initial inoculum. The biomass concentration at startup was about 5 g/L.

Due to the successful cultivation of AGS, removal efficiencies of pollutant in all three reactors were satisfactory. In R2, NLR ranged from 1.50 to 3.30 kg N/(m^3^·d) by reducing the hydraulic retention time from 2.0 to 0.9 h under oxygen-restricted conditions. A very high nitrogen removal rate (NRR) of 2.83 kg N/(m^3^·d) was achieved.

In R3, the removal efficiencies of COD and BOD were observed to be 81–93% and 85–94%, respectively. NH_4_^+^ was nitrated with a removal efficiency of 83–99%, while the resulting nitrate was simultaneously denitrified. Thus, the highest nitrate concentration measured in the effluent was just 4.2 mg/L. The total nitrogen removal efficiency ranged from 52% to 80%, depending on the influent nitrogen concentration (39.3–76.2 mg/L). Phosphate removal efficiencies ranged between 65% and over 99%, depending on the influent phosphate concentration (11.2 to 28.3 mg/L).

The wastewater in reference [6] was dairy waster, the wastewater in reference [8] was municipal wastewater, and the wastewater in reference [20] was high-strength nitrogen wastewater. These naturally generate granules with different microbial compaction, settle ability and generation/maturation time. Because these three kinds of reactors were selected from the different literature, the types of seed sludge were different, and the operational conditions were also quite different. Therefore, we did not pay attention to the comparison of nitrogen removal effects.

### 2.2. Model Geometry for CFD Modeling

Three CFRs were selected to build the CFD model due to their completely different geometries. In Table 1, the volumes of the three reactors are set to be similar in order to compare their hydraulic behavior by CFD modelling.

As shown in Figure 1a, R1 is a reactor with a height of 0.45 m, a length of 0.12 m and a total working volume of 4.3 L, which is used to cultivate aerobic granular sludge. The reactor has an aeration zone and a settling zone, which are separated by a partition. The ratio of the volume of aeration zone to the volume of settling zone is 3:1.

R2 consists of an aerobic volume of 4.5 L, a rectangular internal circulation channel and a settling zone of 1.8 L (Figure 1b).

Different from the vertical structures of the above two CFRs, a rectangular bioreactor R3 was selected with a working volume of 5 L and an aspect ratio of 1.3 (Figure 1c). The reactor has a built-in particle pre-settling zone consisting of a 4-cm-wide sheet positioned at a 60° angle in front of the reactor overflow wall towards the particle settling and recovery zone. A second sheet is also placed at a 60° angle to the wall below the slot where the particles return from the settler, providing shelter from turbulence and facilitating circulation of downward airflow to aid the particle recovery process.

### 2.3. Reactor Meshing

Appendix A shows the meshes of three CFRs. The original grid size of R1 used for the simulation was 35,643 elements, and the radial average side length was 1.0 mm. The original grid size of R2 used for the simulation was 33,947 elements, and the radial average side length was 1.0 mm. The original grid size of R3 used for the simulation was 72,924 elements, and the radial average side length was 1.0 mm.

The top and bottom orifice plates and cylindrical surfaces were modeled as wall boundaries with zero slip, and the outlet was modeled as a pressure outlet. The inlet and air inlet were modeled as velocity inlets. The smooth user-defined function (UDF) was used to model the source of gas and liquid injection points. Source point modeling includes the introduction of mass and momentum source terms into the continuity and momentum equations of the unit where the injection point is located. This allows the use of a coarse grid without the need to refine the orifice plate or the finer details of the injection point of the gas distributor. Appendix A shows the meshes of three CFRs [17].

### 2.4. Mathematical Model

#### 2.4.1. General Information

The numerical simulation of multiphase flow in CFRs system can use a Euler–Lagrange approach. The water and gas are continuous phases, and the particles dispersed in the reactor are discrete phases. The calculation of the continuous phase can be performed by solving the flow field control equation, while the motion and trajectory of the discrete phase need to be calculated by the discrete phase model.

The discrete phase model is actually a model of the interaction between continuous phase and discrete phase matter. In the calculation process with a discrete phase model, the continuous phase flow field is usually calculated first, and then the flow field variables are used to calculate the force received by the discrete phase particles through the discrete phase model, and to determine its trajectory. The discrete phase calculation is carried out under the Lagrangian view; that is, the calculation is performed on a single particle in the calculation process, unlike the continuous phase calculation, which is based on the Euler view, which takes space as the object. The particle size is 0.003 m.

Regarding the turbulent flows, the conservation equations are solved to average time. The time-averaged equations contain additional terms, explaining the transport of mass and momentum by turbulence. In the current simulation, the RNG k–ε model is applied for turbulence closure. The RNG *k*–*ε* model is similar in form to the standard *k*–*ε* model, but it is stronger than the standard *k*–*ε* model in terms of computational functions. This model adds an additional term to the ε equation, which makes it more accurate when calculating flow fields with larger velocity gradients. The rotation effect is taken into account in the model, so the calculation accuracy of the strongly rotating flow is also improved. The standard *k*–*ε* model is a high Reynolds number model, while the RNG *k*–*ε* model can calculate the low Reynolds number effect after proper processing of the near-wall zone. The model contains an analytical formula for calculating the turbulent Prandtl number, unlike the standard *k*–*ε* model, which only uses user-defined constants.

#### 2.4.2. Governing Equations

##### Equation of Particles Motion

The proposed model predicts the trajectories of discrete phase particles by integrating the force balance on the particles, which is written in the Lagrangian reference frame. This force balance equates the inertial forces of the particle with the forces acting on the particle and can be written as Equation (1) [21]
(1)$$m_{p}\left(\frac{du_{p}}{dt}\right)=m_{p}F_{D}\left(u-u_{p}\right)+m_{f}\left(\frac{Du}{Dt}\right)+\frac{1}{2}\left(\frac{Du}{Dt}-\frac{du_{p}}{dt}\right)+\left(m_{p}-m_{f}\right)g+\frac{1}{2}\left(\pi \rho_{p}r^{2}\right)C_{L}LV^{2}L$$
(2)$$F_{D}=\frac{18\mu }{\rho_{p}d_{p}^{2}}\frac{C_{D}Re}{24}\;$$
(3)$$Re=\frac{\rho d_{p}\left|u_{p}-u\right|}{\mu }$$

##### Fluid Phase

By solving the Navier–Stokes equations, the fluid phase is regarded as a continuum. Therefore, in the case of incompressible static turbulence, the mass conservation Equation (4) and momentum conservation Equation (5) can be expressed using the Cartesian tensor representation as follows [21]
(4)$$\frac{\partial u_{i}}{\partial x_{i}}=0\;$$
(5)$$U_{j}\frac{\partial U_{i}}{\partial x_{j}}=-\frac{1}{\rho }\frac{\partial p}{\partial x_{i}}+\frac{\partial }{\partial x_{j}}\left(v\left(\frac{\partial U_{i}}{\partial x_{j}}+\frac{\partial U_{j}}{\partial x_{i}}\right)-\bar{u_{i}^{\prime }u_{j}^{\prime }}\right)\;$$
(6)$$u_{i}=U_{i}+u_{i}^{\prime }\;$$

##### Turbulence

RNG *k*–*ε* involves turbulent kinetic energy *k* and turbulent dissipation rate of turbulence model *ε*. The general transport equation can be described by Equations (7) and (8):(7)$$\frac{\partial }{\partial x_{i}}\left(\rho ku_{i}\right)=\frac{\partial }{\partial x_{j}}\left(a_{k}\mu_{eff}\frac{\partial k}{\partial x_{j}}\right)+G_{k}+G_{b}-\rho \varepsilon \;$$
(8)$$\frac{\partial }{\partial x_{i}}\left(\rho \varepsilon u_{i}\right)=\frac{\partial }{\partial x_{j}}\left(a_{\varepsilon }\mu_{eff}\frac{\partial \varepsilon }{\partial x_{j}}\right)+C_{1\varepsilon }\frac{\varepsilon }{k}\left(G_{k}+C_{3\varepsilon }G_{b}\right)-C_{2\varepsilon }\rho \frac{\varepsilon^{2}}{k}-R_{\varepsilon }\;$$

##### Discrete Phase Model

In this study, the discrete phase model is used to simulate the motion of solid particles. Since the content of aerobic granular sludge in water is very small (less than 10% volume fraction), the interaction between particles and the influence of particles on the flow field can be ignored in the simulation. The motion equation of a solid particle is shown in Equation (9) [22]
(9)$$\frac{du_{p}}{dt}=F_{D}\left(u-u_{p}\right)+\frac{g_{x}\left(\rho_{p}-\rho \right)}{\rho_{p}}+F_{x}\;$$

### 2.5. Boundary Conditions

At inlets, a velocity inlet boundary condition is used. This type of boundary condition is suitable for incompressible flows, and if used for compressible flows, it will lead to non-physical results. The initial values of *k* and ϵ are calculated from Equations (10) and (11), respectively, assuming values of $T_{u}^{2}\;$= 0.05, *C_μ_* = 0.09 and turbulence length scale (*L_u_*) of 0.5 times the inflow radius to the inlet baffle. Turbulent viscosity *v_t_* is computed automatically by OpenFOAM^®^.
(10)$$K=1.5{\left(T_{u}v_{m}\right)}^{2}$$
(11)$$\varepsilon =\frac{C_{\mu }k^{\frac{3}{2}}}{L_{u}}$$

For the outlet, pressure outlet boundary conditions are applied. With this type of boundary condition, all the flow quantities at the outlets are inferred from the flow in the inner domain. It is also possible to specify a set of “backflow” conditions that allow for reverse flow at the pressure outlet boundary during the solution process. Symmetric boundary conditions are used in free water surface. The solid boundary is specified as a fixed wall with no-slip shear condition.

For the discrete phase model, the additional boundary conditions are as follows: for the velocity inlet and pressure outlet, the “escape” condition is specified. This means that when the particle encounters the boundary, it is reported as having escaped, which is assumed in all flow boundaries. The “reflect” condition is specified near the solid boundary. This means that the particle bounces off the boundary in question and its momentum changes, as defined by the coefficient of recovery. Finally, use the “trapped” condition at the bottom of the tank. This means that the trajectory calculation is terminated, and when the particle encounters the proposed boundary, its fate is recorded as being captured [21].

### 2.6. Simulation

The steady state flow field is obtained by solving the continuous phase flow field without particles. The convergent solution is defined as the solution with normalized residuals less than 10^−4^ for all variables. All CFD simulations were performed at a constant time step of Δt = 0.1 s, with a time average of another 10 s during the run after the simulation ran a clock time of 100 s and reached a fully developed transient flow.

### 2.7. CFD Model Validation

To determine the accuracy of CFD model, the liquid flow velocity at various locations in R3 was measured, which was compared with the simulated data to verify the accuracy of the model. In the experiment, a portable flow velocity meter (LS300A-Nanjing) was used to measure the liquid flow velocity in the physical reactors. The sampling points were located on the lines parallel to the *Y*-axis at the middle of the secondary settling zone and the settling zone of R3, respectively.

### 2.8. Retention Time Distribution Analysis

A 200 g/L concentrated NaCl solution was injected in pulses at the inlet of R3. The electronic conductivity (EC) of the effluent was recorded every five minutes by an EC meter (WTW 3310). Calibration curves were drawn by measuring EC values of standard NaCl solutions (concentration from 0 to 1 g/L).

Breakthrough curves were obtained by the retention time distribution E(t) in Equation (12)
(12)$$E(t)=\frac{C(t)Q(t)}{M_{\mathrm{tracer}}}$$
where M_tracer_ is the total mass of tracer injected (g), C(t) is tracer concentration (g/L) and Q(t) is water flow rate at the outlet (L/min).

The actual hydraulic retention time t_actual_ was calculated by the first order moment analysis in Equation (13) and the variance σ^2^ was induced by the second order moment analysis of BTCs in Equation (14)
(13)$$t_{\mathrm{actual}}=\int_{0}^{\infty }E\left(t\right)tdt$$
(14)$$\sigma^{2}=\int_{0}^{\infty }E\left(t\right){\left(t-t_{\mathrm{actual}}\right)}^{2}dt\;$$

The theoretical retention time (τ) was calculated by Equation (15)
(15)$$\tau =\frac{V\mathrm{void}}{Q}$$
where V_void_ is total void volume in reactors.

If t_actual_ < τ, there was a dead volume; if t_actual_ > τ, a short circuit phenomenon occurred [23].

Parameter e_v_ was introduced to describe the hydrodynamic behavior by Equation (16)
(16)$$e_{v}=\frac{t_{\mathrm{actual}}}{\tau }$$

The reactor dispersion index (MDI) was calculated by Equation (17)
(17)$$\mathrm{MDI}=\frac{t_{90}}{t_{10}}$$
where t_10_ and t_90_ are the 10% and 90% mass passing through the outlet, respectively.

## 3. Results and Discussions

### 3.1. Validation of CFD Model

R3 was selected to validate the accuracy of the CFD modelling. The variation in the liquid velocity along the axial direction of R3 under different baffle angles is shown in Figure 2. It can be seen that the model output was basically consistent with the experimental results with a difference less than 10%, indicating that the model can be used to predict the flow characteristics in the reactor under different baffle angles. In Figure 2a, with the increase in the distance in the *Y*-axis direction, the liquid flow velocity gradually increased, which was due to the existence of dead volume at the bottom of the secondary settling zone. This facilitated the settling of the particles. Conversely, the liquid flow velocity first increased and then decreased along with the increase in the *Y*-axis direction (Figure 2b). In particular, the liquid flow velocity in the three reactors reached the highest at 0.02–0.04 m. This was because on the right side of the 0.02–0.04 m was the water inlet, and the pulse effect of the water flow increased the liquid flow velocity.

### 3.2. Comparison of Hydrodynamic Behavior in Three CFRs

#### 3.2.1. Fluid Behavior

The distributions of the velocity contour and streamline of liquid fluid at the vertical sections (Z = 0) of R1, R2 and R3 are shown in Figure 3. It can be seen that R1 and R2 were vertical flow reactors. The difference was the position of the water inlet baffle and the shape of the settling zone. The inlet water of R1 flowed directly from the top to the bottom, while the inlet water of R2 was divided into two by the baffle, and most of the water flowed up and then down. R3 was a cross-flow reactor.

Regarding the contours of three CFRs (Figure 3a–c), the water velocity at the bottom of the reactor was large and a turbulence with a large diameter was formed both in R1 and R3. This may be caused by aeration at the bottom of the reactor. Afterwards, the water velocity around both the baffle and near the wall of the reactor was extremely low, although the inlet water direction was different. Specifically, the water velocity in the granules’ settling and recovery zone, secondary settling zone and overflow zone of R3 was relatively small, 0.3–0.5 m/s. However, the contour of R2 seemed to be quite different. It was observed that the water velocity in the first half zone of the reactor was small, and became larger in the settling zone. The reason could be the large inclination angle at the settling zone.

The distributions of the transient liquid velocity streamline of R1 to R3 are illustrated in Figure 3d–f, respectively. In R1, turbulence was observed in the vertical direction of the aeration zone, with the water descending in the center of the reactor and ascending along the wall. A turbulent flow with a diameter of one-half the height of the reactor was formed at the bottom of the reactor. Before the outlet, the water flowed to the settlement zone, forming several small turbulences at the vertical height of the settlement zone. Unlike R1, R2 owned a greater number of baffles and a settling zone with a large inclination angle. Thus, no obvious turbulence was found with only little turbulence at the inlet zone.

Different from R1 and R2, R3 owned a horizontal water flow behavior. As presented in Figure 3f, a small turbulence was formed at the inlet. Then the water flowed upwards, forming two symmetrical turbulences. According to Tarpagkou [21], the large number and small radius of turbulences signify that the degree of liquid phase mixing is better. Part of the water bypassed the baffle in the granules’ pre-settling zone and flowed to the water outlet, and the other part flowed through the granules’ settling and recovery zone and formed a vortex. After that, the water flowed through the secondary settling zone to form a turbulent flow with a larger diameter and finally exited the reactor.

Overall, the vertical flow mode had a good particle separation effect in R1 and R2, but the turbulent flow effect was poor and the water shearing force was small, which is not conducive to the cultivation of AGS [10]. Conversely, there was more asymmetric turbulence distributed in R3, allowing the liquid phase mixing in the entire reactor to be relatively uniform. A similar flow pattern of turbulence also appeared in other physical phenomena, such as aeromechanics [24] and biomechanics [25,26].

#### 3.2.2. Granular Separation

The particle distribution contour plots of R1 to R3 are illustrated in Figure 4a–c, respectively. It can be observed that the particles’ concentration was always maintained at a high level in the aeration zone with almost no particles settling in the upper part of the outlet and the settling zone in R1. The particle separation effect was acceptable, whereas most of the particles were kept at the bottom and in the settling zone of R2, probably due to the function of the baffle and inclination angle. Moreover, the concentration of particles at the top of the rectangular internal recycling channel was relatively large. It may be that the particles flowed from the water inlet to the upper part of the reactor, and some particles were left above the baffle in the rectangular internal recycling channel. The particles’ concentration near the outlet was zero, which indicates that the particle sedimentation effect was good.

Regarding R3, the particle concentration at the bottom was relatively large. The particles observed in the secondary settling zone indicate that some parts of the particles passed through the granular pre-settling zone, also highlighting the necessity for a secondary settling zone. Similar to the former two CFRs, there were no particles in the overflow zone and the outlet, indicating that the particle separation effect was also acceptable.

### 3.3. Structural Optimization of R3

#### 3.3.1. Optimizing Baffle Distances

According to the comparison of the results of the three CFRs, R3 was selected as the optimal one. Furthermore, it is essential to optimize its design for a better hydrodynamic behavior and granular separation performance. Firstly, baffle distances of 30, 40 and 50 mm were evaluated to compare their fluid behavior and particle separation performance. The angle of the baffle was fixed to be 60°.

##### Judging from Fluid Behavior

Figure 5b,e show the water velocity contour plots and water velocity streamline of the original R3 with a baffle distance of 40 mm. The other four figures (Figure 5a,c,d,f) present the results of R3 with a baffle distance of 30 and 50 mm. The water velocity was always kept highest at the bottom of the reactor due to the aeration. When water reached the top of the reactor, it began to diverge. Specifically, one part flowed to the settling zone, one part flowed to the secondary settling zone and the remaining part flowed directly out of the reactor. There was a circular velocity area at the top of the settling zone and the secondary settling zone.

The difference among the treatments of the three baffle distances belonged to the turbulence pattern in the aeration zone and the water velocity in the settling zone, which increased with the augmentation of the baffle distance. For instance, the water flow velocity in the upper part of the particle settling zone and the secondary settling zone varied between 0.02–0.06 m/s. The water flow velocity in the overflow zone was relatively high.

Through the above comparative analysis, it can be concluded that the reactor with a baffle distance of 40 mm had the most abundant turbulence and the ideal liquid phase mixing pattern.

##### Judging from Granular Separation

Figure 5g–i show the particle distribution contours of baffle distances of 30, 40 and 50 mm, respectively. It can be observed that the particle concentration was relatively large at the bottom of the reactor, and it gradually decreased upwards. The particle concentration in the upper half of the reactor was almost zero. Along with the increase in the baffle distance, the change in the particles’ concentration in the aeration zone was not significant. However, their concentration in the settling zone augmented along with the increase in distance. For example, for a baffle of 50 mm, in the secondary settling zone, a large number of particles settled, and the particles’ concentration in the overflow zone was larger. Finally, the particles flowed out of the reactor at the outlet, which proves that the particles were not well separated. The reason could be attributed to the overlarge flow channel at a baffle distance of 50 mm that brought some particles into the settling zone.

In order to analyze the sedimentation effect of the particles in the reactor more accurately, two sections parallel to the YZ plane in the positive direction of the X-axis were selected to study the profile of the particles’ concentration: the pre-sedimentation zone and secondary settling zone, and a straight line perpendicular to the XY plane in the middle of each plane was selected to study the particles’ concentration profile (Figure 6).

It can be seen that the particles’ mass concentration in the pre-sedimentation zone of the reactor was much greater than that in the secondary sedimentation zone. Furthermore, along with the increase in baffle distance, the particles’ mass concentration increased correspondingly. The declining trend of the particles’ mass concentration in the three reactors was roughly the same, decreasing significantly from 0 to 0.016 m. The particles’ mass concentration in the two reactors with a baffle distance of 30 mm and 40 mm had similar decreasing trends. However, the baffle distance of 50 mm possessed a large concentration of particles at 0.016 m, which is consistent with the particle distribution contour plots. Finally, the particles’ concentration declined to close to zero at 0.110 m for all three baffle distances.

Figure 6b presents the particle concentration in the secondary sedimentation zone of the reactor. Obviously, the concentration for the baffle distance of 30 mm was always zero, showing the first sedimentation zone played a perfect separation function. The decreasing trend of the baffle distances of 40 and 50 mm was basically consistent before 0.080 mm. Afterwards, it took a larger distance for the baffle distance of 50 mm to reach the highest particle concentration at 0.114 m. Then, the particles’ concentration gradually decreased. The reactor with a baffle distance of 40 mm had the maximum particles’ mass concentration at 0.088 m, and then the particles’ mass concentration dropped to zero at 0.114 m.

Through the analysis and comparison of the fluid path and particle separation effect of the reactors with different baffle distances, it is concluded that the reactor with a baffle distance of 40 mm had relatively uniform liquid phase mixing and better particle separation effects. The angle of the baffles was further optimized.

#### 3.3.2. Optimizing Baffle Angles

##### Judging from Fluid Behavior

The baffle angles of 30°, 45°, 60° and 75° were selected to compare and analyze their fluid paths and particle separation effects to obtain the best baffle angle. Figure 7a–h show the water velocity contour plots and water velocity streamline of the reactors with the baffle angles of 30°, 45°, 60° and 75° at the baffle distance of 40 mm. It was observed that the high water velocity occurred at the inlet, and the low water velocity existed in the settling zone and the secondary settling zone.

The water velocity contour plots and water velocity streamline of the reactors with the baffle angles of 30° and 45° were approximately the same. A small turbulence formed at the air diffuser. Then, the water flowed upward, forming a turbulent flow with a larger diameter below the pre-sedimentation zone. The other part of the water flowed through the settling zone and then flowed away in two directions. Part of the flow flowed to the secondary settling zone and then to the outlet, and the other part flowed back to the upper part of the pre-settling zone to form a turbulent flow with a small diameter.

A different situation occurred at the angle of 60° (Figure 7g). It was observed that the diameter of the turbulence below the pre-sedimentation zone of the reactor became small as the angle of the baffle increased. Meanwhile, two equal-sized turbulences were formed in the secondary sedimentation zone of the reactor, and then flowed out of the reactor. Furthermore, when the angle was increased to 75°, it could be seen that a turbulent flow with a small diameter appeared under the baffle in the settling zone (Figure 7h). Then part of the water flowed through the pre-sedimentation zone and part of the water flowed through the settling zone, and then it flowed out of the reactor through the secondary settling zone.

##### Judging from Granular Separation

Figure 7i–l present the particle distribution contour plots of the baffle angles of 30°, 45°, 60° and 75° with the baffle distance of 40 mm. It was observed that the particles at the bottom of the four reactors had an abundant number, and the particles’ concentration in the pre-sedimentation zone was almost zero. However, particles were still found to be settled in the secondary sedimentation zone, which proved that particles passed through the pre-sedimentation zone. The difference was that the reactor with a baffle angle of 30° had particle sedimentation in the overflow zone, but the other three reactors did not, indicating that the particles in the other three reactors were well separated. At the same time, as the angle of the baffle increased, the particles’ mass concentration in the secondary sedimentation zone gradually decreased, and the particles were accumulated in the secondary sedimentation zone at the baffle angles of 60° and 75°.

In order to analyze the sedimentation effect of particles in the reactor more accurately, two sections parallel to the YZ plane in the positive direction of the X-axis were selected to study the profile of the particles’ concentration: the pre-sedimentation zone and secondary settling zone, and a straight line perpendicular to the XY plane in the middle of each plane was selected to study the particle’s concentration profile Figure 8.

In Figure 8a, the particles’ mass concentration gradually decreased as the angle of the baffle increased. The particles’ mass concentration in the pre-sedimentation zone of all the reactors gradually decreased from the bottom up, and it dropped to zero at 0.031 m. Different phenomenon occurred in the secondary settlement zone. Figure 8b indicates that except for the reactor with a baffle angle of 60°, the particles’ mass concentration in the secondary settlement zone of the other three reactors gradually decreased. In the reactor with a baffle angle of 60°, the particles’ mass concentration gradually decreased at a position from 0.069 m to 0.079 m, and then the particles’ concentration increased. Its concentration reached the highest value at 0.0880 m, and then dropped to zero at 0.114 m.

Through the analysis and comparison of the flow path and particle separation effect of reactors with different baffle angles, it was concluded that the liquid phase mixing of the reactor with a baffle angle of 60° was more uniform and the particle separation effect was better.

### 3.4. Hydraulic Parameters by RTD Analysis

Analytical methods related to the residence time distribution are very useful to characterize the hydraulics and macroscopic mixing in an integrated way with no spatial information, but they can determine reactor malfunctioning flows such as short-circuiting or dead volume [27].

Figure 9 shows the tracer E(t)-time experimental data obtained at different baffle angles and at different baffle distances from the reactor corresponding to the outlet in the aeration cycle for 10 h. From Figure 9a, it can be observed that the three reactors with different baffle angles corresponded to similar hydrodynamic performances, noting that the global flow behavior was related to the mixed flow and that the maximum E(t) of the tracer was reached in a similar time (after 10 min). Then E(t) decayed exponentially, corresponding to a continuous stirred tank reactor. It is confirmed that the bulk fluid behavior corresponded to a perfectly mixed flow, according to the low MDI values in Table 2 [28]. The tracer concentration showed slow circulation inside the reactor with baffle angles of 45° and 75°, which indicated insufficient mixing. This was due to the change in the water flow path caused by different baffle angles in the reactors. From Figure 9b, it can be observed that the three reactors with different baffle distances corresponded to similar hydrodynamic performances, noting that the global flow behavior was related to the mixed flow. The tracer concentration showed slow circulation inside the reactor with a baffle distance of 30 mm, which indicated insufficient mixing.

Estimated hydraulic parameters were presented in Table 2. The actual HRT was less than the nominal HRT in all the tracer tests, which indicates that there was a certain volume unavailable to water flow. The reason could be that water always flowed through preferential paths inside the R3. In particular, the reactor with the baffle distance of 40 mm had the largest effective volume ratio, and the reactor with the baffle angle of 60° had the largest effective volume ratio. This shows that a baffle distance of 40 mm and a baffle angle of 60° in a reactor were the two preferential parameters, which is in accordance with CFD modelling.

## 4. Conclusions

This study clarifies the granulation efficiency, hydrodynamic behavior and particle separation of three different CFRs. The results show that all three CFRs were able to realize the granulation with the satisfied nitrogen removal efficiencies, and R3 owned the shortest granulating time. The established CFD model was able to explain the hydrodynamic behavior. Regarding the particles’ separation efficiency, there was no great difference between the three CFRs, but R3 performed the best from the perspective of hydrodynamic behavior due to its high turbulence and high shear force. In addition, the baffle distance and baffle angle of R3 were suggested to be 40 mm and 60°, respectively.

## Figures and Tables

**Figure 1 ijerph-19-08306-f001:**
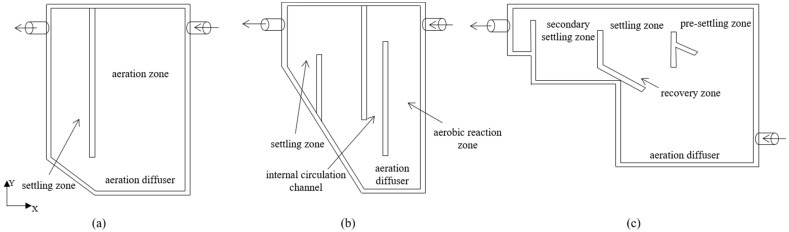
Geometry of three CFRs: R1 (**a**), R2 (**b**) and R3 (**c**).

**Figure 2 ijerph-19-08306-f002:**
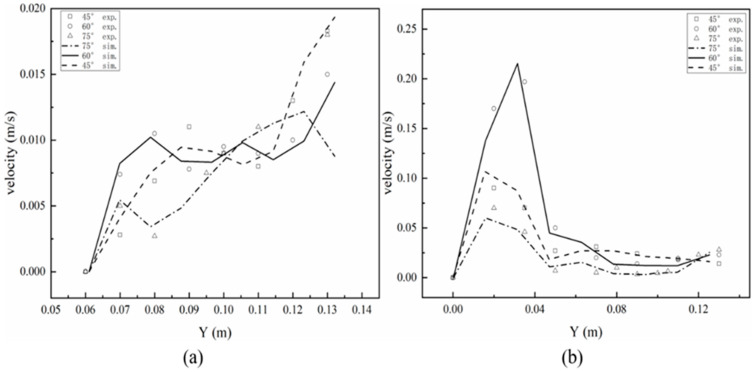
Comparison of the experimental and simulated liquid velocities along the *Y*-axis of the reactor under different baffle angles in the middle of the secondary settling zone (**a**) and settling zone (**b**) in R3.

**Figure 3 ijerph-19-08306-f003:**
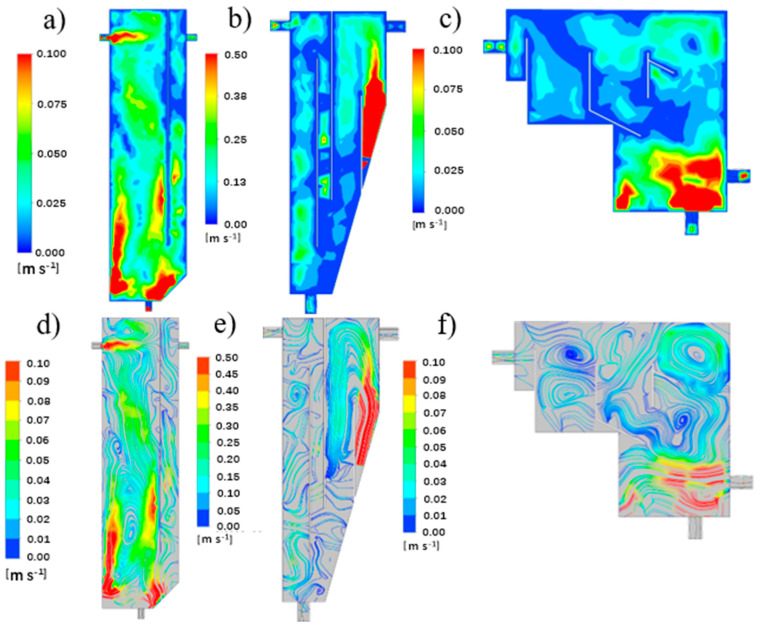
The water velocity contour and liquid velocity streamline of R1 (**a**,**d**), R2 (**b**,**e**) and R3 (**c**,**f**).

**Figure 4 ijerph-19-08306-f004:**
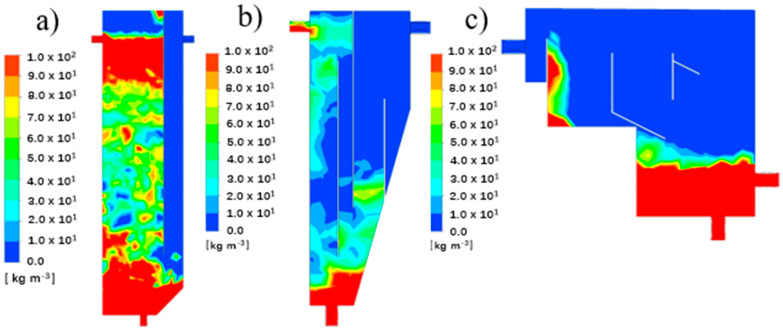
The particle distribution contour plots of R1 (**a**), R2 (**b**) and R3 (**c**).

**Figure 5 ijerph-19-08306-f005:**
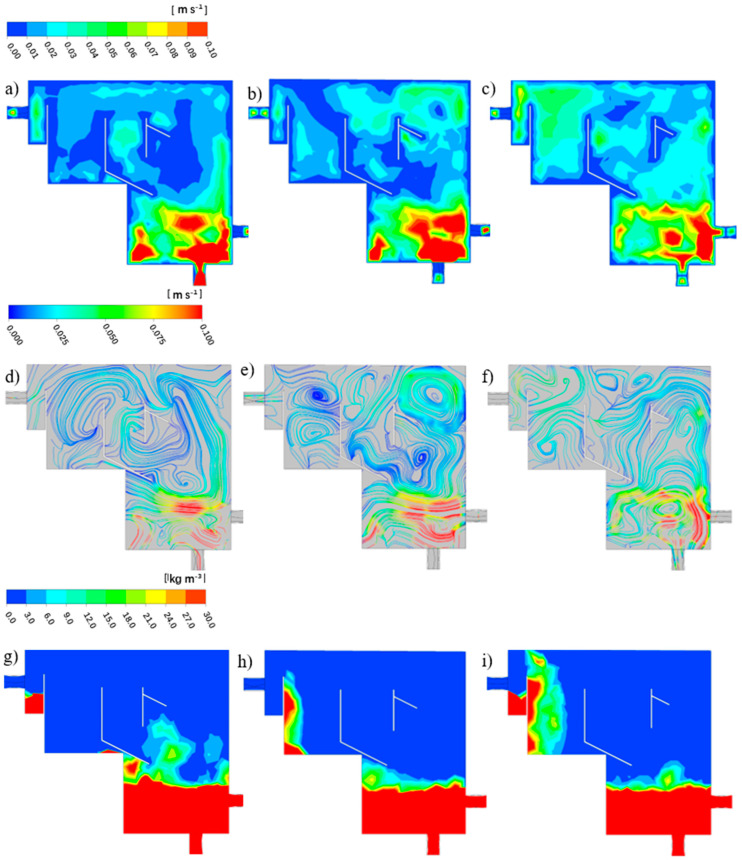
The water velocity contour plots, liquid velocity streamline and the particle distribution contour plots of the four types of R3 with the baffle distance of 30 mm (**a**,**d**,**g**),40 mm (**b**,**e**,**h**) and 50 mm (**c**,**f**,**i**).

**Figure 6 ijerph-19-08306-f006:**
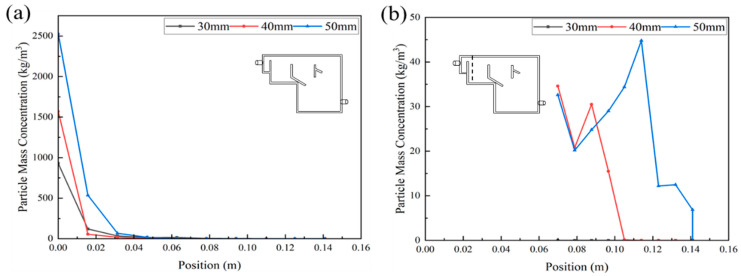
The change in particles’ mass concentration in the vertical direction of the pre-sedimentation zone (**a**) and secondary settling zone (**b**) of the reactor with different baffle distances.

**Figure 7 ijerph-19-08306-f007:**
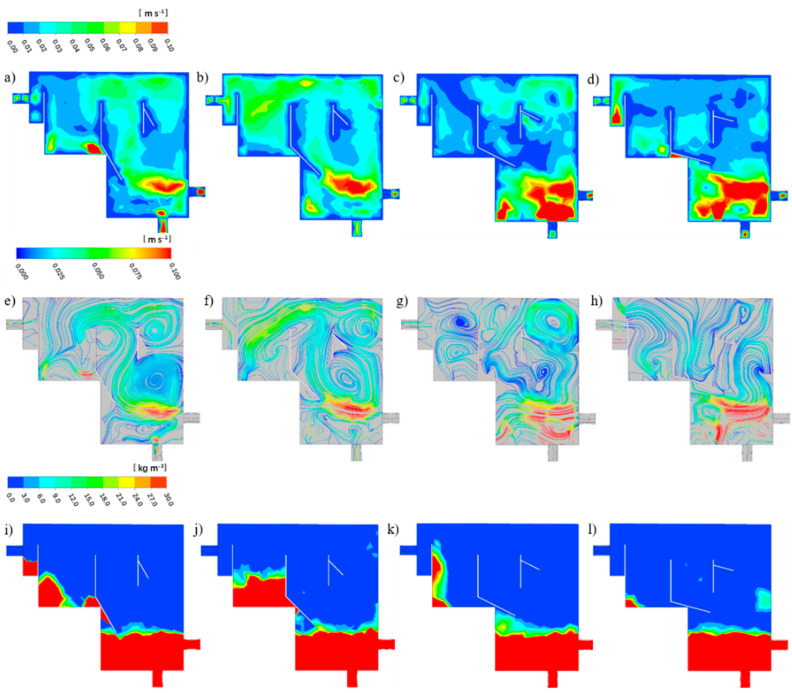
The water velocity contour plots, liquid velocity streamline and the particle distribution contour plots of R3 with the baffle angle of 30° (**a**,**e**,**i**), 45° (**b**,**f**,**j**), 60° (**c**,**g**,**k**) and 75° (**d**,**h**,**l**).

**Figure 8 ijerph-19-08306-f008:**
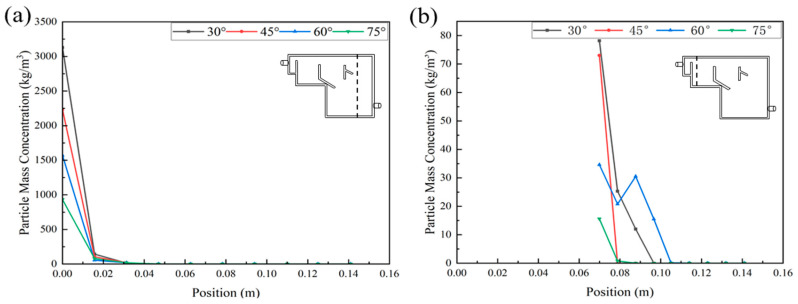
The change in particle concentration in the vertical direction of the pre-sedimentation zone (**a**) and secondary settling zone (**b**) of the reactor with different baffle angles.

**Figure 9 ijerph-19-08306-f009:**
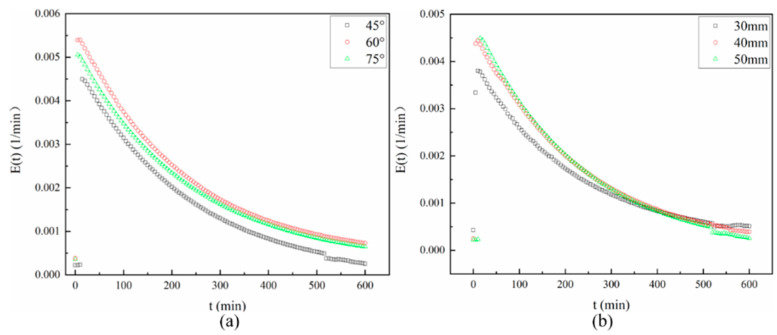
E(t) versus time tracer response curves obtained at the reactor outlet for different baffle angles (**a**) and different baffle distances (**b**).

**Table 1 ijerph-19-08306-t001:** Comparison of operational parameters and granulating efficiency in three CFRs.

	V	HRT	Baffle Number	Q_water_	Q_gas_	Granulation Time	SVI_5_	Granular Diameter	Ref.
	(L)	(hour)		(L/h)	(L/min)	(days)	(mL/g)	(mm)	
R1	4.3	0.55	1	7.8	0.3	40	44–61	0.5–2.0	[8]
R2	1.5	0.9	3	1.7	0.9	33	87	1.2	[20]
R3	10	10	3	1	6–8	14	/	2	[6]

HRT: hydraulic retention time; SVI_5_: 5-min sludge volume index.

**Table 2 ijerph-19-08306-t002:** Variation in hydraulic parameters of R3 under different baffle distances and angles.

Case	t_actual_	τ	e_V_	σ^2^	t_10_	t_90_	MDI
	min	min			g	g	
Distance 30 mm	118	210	0.56	10,140	1.29	0.22	0.17
Distance 40 mm	134	210	0.64	10,519	1.52	0.21	0.14
Distance 50 mm	133	210	0.64	9979	1.56	0.15	0.1
Angle 45°	134	210	0.64	10,519	1.52	0.21	0.14
Angle 60°	172	2108	0.82	20,026	1.85	0.35	0.19
Angle 75°	160	210	0.77	18,659	1.7	0.31	0.18

t_actual:_ actual hydraulic retention time; τ: theoretical retention time; e_v_: effective volume; t_10_:10% mass passing through the outlet; t_90_: 90% mass passing through the outlet; MDI: reactor dispersion index.

## Data Availability

Not applicable.

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
