# Peer review of "Comparison and Optimization of Continuous Flow Reactors for Aerobic Granule Sludge Cultivation from the Perspective of Hydrodynamic Behavior"

_ijerph, 2022, doi:10.3390/ijerph19148306_

Round 1

Reviewer 1 Report

This study aimed to investigate the use of computational fluid dynamics (CFD) to optimise the generation and stable operation of aerobic granular sludge (AGS) in a continuous flow reactor (CFR). The study utilises the data presented in three previously published papers (reference 6, 8 and 20) to generate the CFD models, however this is not clear until Table 1 is reviewed.

The papers used to generate the data found in Table 1 derived AGS from different wastewater streams, however this is not noted (6: Dairy Waste, 8: Municipal Wastewater, 20: High strength nitrogen wastewater, AGS optimised to achieve CANON-process). This naturally generates granules with different microbial compaction, settleability and generation/maturation time.

There are many additional studies into AGS development in CFR operation which have not been investigated or mentioned. There is potential these studies would have overcome the differences in wastewater matrix and AGS development times. The paper does not mention how this can be applied in field scale or if there is a prediction of changed CFD outcomes when scaled up.

Line 24: Requires temporal duration, 0.82 d (days) or 0.82 hrs (hours)

Line 32: Important to clarify here, either ammonia and total nitrogen or nitrogen as ammonia.

Line 42: no need to capitalise granulation

Line 42 – 44 – statement not clear. Do you propose that a short period of time taken to establish mature granular sludge is preferred as this enable the biomass to tolerate potentially harmful pollutants?

Line 50: SBRs are commonly used for wastewater treatment due to the compact nature, the issue with SBRs is ability to treat high volumes without requiring a large number of units. Many SBR based WWTPs already operate with aerobic granular sludge with a high degree of success globally.

Section 2.1: title misleading. Majority of this can be condensed as the test replicates the data found in Table 1.

Line 84: Incomplete sentence

Line 96: First use of NLR?

Line 105: The reactor volumes in the literature were not the same, 4.3, 1.5 and 10L respectively. The data presented in Table 1 contradicts this statement for all parameters mentioned. Additionally, Line 107 – 119 contradict Line 105 further.

Section 2.2: Were these CFR models taken directly from the referenced papers?

Line 214: was the study in reference 20 conducted with this paper in mind? or are the points used to validate the CFD model mentioned in section 2.7 attained from the paper? If so, reference the paper as the source of the flow data.

Line 243: was shown, should be is shown

Line 330: needs to clearly indicate on Figure 5 the baffle movement

Section 4: While difficult to assess, there are no comments as to the impact of nutrient removal performance of the changes in baffle.

Author Response

For details, please see the word file.

Reviewer 2 Report

The paper is very interesting. The research is well thought out although reading the article is difficult. The difficulty is that in Fig. 1, the appearance of the reactors is flat and it is not known what it looks like spatially, and the vortices formed during the flow of wastewater are spatial. Therefore, not all explanations of phenomena are easy to imagine. 

I have 2 questions. It would be good to include comments on this in the text.

1. The main task of biological reactors, whether flow or sequential, is wastewater treatment. There is no comment on whether the wastewater treatment was effective under the conditions of the assumed research?

2. The mathematical modeling of hydraulic phenomena had to refer to a specific size (diameter of granules) or a range of their size. What size of granules was involved in the flow hydraulics modeling?

Additional remarks

3. p. 9, 290 - is Fan and Yuan [23] - paper [23] is from other authors

2. No paper citation [24]

Author Response

For details, please see the word file.

Reviewer 3 Report

The present work reports on comparison and optimization of continuous-flow reactors for aerobic granule sludge cultivation from the perspectives of hydrodynamic behaviour. The work is highly actual nowadays. Nevertheless, points below should be addressed prior to the submission:

 Line 42:  It is: „ The Granulation ….”; It should be “The granulation……”

Table 2: The lack the description of acronyms:  HRT, SVI5.

Author Response

For details, please see the word file.

This manuscript is a resubmission of an earlier submission. The following is a list of the peer review reports and author responses from that submission.

Round 1

Reviewer 1 Report

The authors studied the effect that different geometrical configuration on aerobic granular sludge reactors have over the granulation and characteristics of the biomass.

I think that the research is very novel and significant in the field of wastewater treatment.

Moreover, the scientific soundness seems very appropriate to me, but I am not an expert in modelling, so I'll have others to evaluate that with a more accurate opinion than mine.

I believe that the writing is good, the results clearly presented, and that the paper is easy to follow when reading.

Have two minor suggestions for the authors to strengthen the scope of this manuscript.

1) The authors study the granulation efficiency (granulation speed) of three technological configurations. This is a good approach, not all configurations available can be tested. However, I would like to know why these three specific configurations were used. Is there any literature about them? I would suggest the authors to provide more info on their choice of these configurations in the materials and methods section.

2) The authors develop a preliminary study on the three systems to see which provides faster granulation of biomass, then focus on that particular configuration for more detailed studies. I think that the authors move in the right direction when focusing on the configuration that provides shorter granulation time. However, why is faster granulation desired? This question surely is very clear in the mind of people that are familiar with the technology, but maybe not for the wider public. I would suggest the authors to provide insights into why shorter granulation time is preferred. A couple lines in the introduction section would be good for this.

Reviewer 2 Report

In the manuscript modelling of continuous-flow reactors for cultivation of aerobic granular sludge (AGS) was performed. The subject is important because identification of operational parameters and reactor geometry that support granulation in continuous-flow reactors will create an opportunity for broader application of AGS. The way the experiment was planned did not allow to fulfill the goal of the study.

From the abstract it can be concluded that reactor geometry was the main factor that was investigated followed by retention time. The description of reactors’ operation indicated that they differed not only in geometry but also in crucial operational parameters such as HRT or aeration intensity. In my opinion it makes it impossible to draw proper conclusions - we cannot state that the granulation is the effect of reactor geometry and not the intensity of e.g. aeration. How can we compare R1 with R3 if not only reactor configuration but also aeration intensities are different? The authors stress in the conclusion section that R3 performed best due to high shear force – not the reactor geometry. In fact, there were huge differences in the air flow – over 20 times higher rate was applied in R3 than in R1. Aeration intensity affects strongly granulation changing microbial behavior e.g. extracellular polymer excretion. Thus, in my opinion the whole modeling of hydrodynamic behavior should be only focused on R3.

I could not assess the meshes in CRFs because there were no supplementary materials available.

The abstract is unclear. The differences between the investigated CFRs should be specified. There is a phrase „these optimal parameters” – which in particular? Retention time distribution was not explained at all. Baffle angle is not an operational parameter.

Introduction: “large municipal sewage”, “large amount of water treatment” – unclear meaning.

Point 2.2- lines 89-93 repeated in lies 101-105.

L 209 – what kind of filters?

How was the granulation time estimated in this study? How was the efficiency of granulation estimated? This was the goal of the study as expressed in the manuscript title.

Whose results are discussed in point 3.1 – the authors from references 7, 21 and 6? Are there any results from this study regarding e.g. granule diameters?

Reviewer 3 Report

The authors have compared the granulation efficiency and the hydrodynamic behavior of three reactors. After choosing the best one, they compared again these parameters modifying the baffle distance and the baffle angle.

Please address the following issues:

  • There is no comment on how granulation is produced: kind of sludge inoculated; volume inoculated in each reactor...
  • What was the biomass concentration (in terms of Mix Liquor Suspended Solids) that there was in each reactor?
  • With the aim to compare some reactors, it would be preferable that the volumes were the same, or at least the hydraulic retention time. Why is this parameter different between the reactors? How does it affect the conclusions?
  • In Table 1, I understand that the parameters are from your own reactors, why are there references?
  • It is not clear how settled granules (in the settling zone of R2, secondary settling zone of R3 and the last small chamber of R3) can return to the aeration zone.
  • Coordinate axes (X, Y, Z) in Figure 1 could be removed, because the reactors are drawn in two dimensions and it is not possible to observe the reactor in the Z axe.
  • Please, revise and check carefully the final version of the manuscript, since there are some paragraphs duplicated (lines 89-101 and 101-113). Also, there are unclear sentences such as "nitridation granular sludge" in line 228. In Table 2, row 7, column 3. Even in the title "aerobic granulAR sludge". Please, revise completely the manuscript.

Round 2

Reviewer 3 Report

Thank you for your comments. Some aspects of the manuscript still need to be clarified.

  • Along the manuscript it is not clear if you had only simulated reactors or if there is any physical reactor. I imagine that you have to have a physical reactor to validate the model, as you comment in sections 2.6 and 3.2, but that issue is not clear. Could you clarify this aspect?
  • In section 3.1, you compared three reactors of the bibliography, however those reactors are not compared with the data of your model, so it seems a revision article and not an original article. Please, comment and discuss these data with yours.
  • In Table 1, references 7 and 21 do not correspond with data of the table. Reference 7 is about SBR (not CFR) and it has other parameters. Reference 21 is not about aerobic granular sludge.